# The role of the arts in enhancing data literacy: A scoping review protocol

**Ailish Hannigan** [1,2] *, **Fran Garry** [2,3], **Conor Byrne** [1], **Helen Phelan** [2,3]

**1** School of Medicine, University of Limerick, Limerick, Ireland, **2** Health Research Institute, University of Limerick, Limerick, Ireland, **3** Irish World Academy of Music and Dance, University of Limerick, Limerick, Ireland

* ailish.hannigan@ul.ie

**Data Availability Statement:** No datasets were generated or analysed during the current study. All relevant data from this study will be made available upon study completion.

## Abstract

Data literacy has been defined as "the ability to read, work with, analyze and argue with data". The United Nations has highlighted a growing risk of inequality for people excluded from the new world of data by lack of education, language, poverty, and discrimination and has called for the development of data literacy at all levels of society. Responses to data are shaped by personal, social and cultural influences, as well as by trust in the source. The arts can play an important role in regulating our responses to information and increasing accessibility, engagement and sense-making of data. However, to our knowledge, to date, there has been no comprehensive review of publications on the role of the arts in the context of data literacy. This paper presents a protocol and a methodological framework to perform a scoping review to identify and map the available evidence for the role of the arts in enhancing data literacy. The review aims to provide an overview of research over the past twenty years to develop a clearer understanding of (a) which art forms are represented in the literature (b) which population groups and settings are identified (c) and the rationale for using the arts to enhance data literacy.

## Introduction

Technological advances have led to increased volumes of data being created and consumed in many areas of our lives including education, work, health and social interactions. This datafication of society has led to calls for the development of data literacy to inform and empower people living in the digital age [1]. There are multiple definitions of data literacy, depending on the field of study and context. Bhargava et al. [2] define it as "the ability to read, work with, analyze and argue with data" (p. 198). Gray et al. refer to the overlap with statistical literacy, where a set of skills are actively used to understand statistical information, but also include the ability to use data critically, to make ethical data decisions and address trust in data sources [3]. The ability to process and use information in different formats, assessing its credibility and relevance, is also considered a key component of health literacy [4].

The United Nations has highlighted a growing risk of inequality with many people excluded from the new world of data and information by lack of education, language, poverty, and

**Funding:** This work has been supported by the Irish Research Council https://research.ie/ under grant number COALESCE/2022/1664 (HP, AH). The funders had and will not have a role in study design, data collection and analysis, decision to publish, or preparation of the manuscript. There was no additional external funding received for this study.

**Competing interests:** The authors have declared that no competing interests exist.

discrimination and has called for the development of data literacy at all levels of society to fully implement the Sustainable Development Goals [5]. A lack of data literacy can limit people's ability to identify mis- and disinformation, exposing them to risks in many areas of their lives [6]. Mis-information is "when false information is shared, but no harm is meant" (p. 5) [7]. Disinformation is "false information knowingly shared to cause harm" (p.5) [7]. Both can have negative impacts on people's health and wellbeing and increase the risk of creating conflict and division in society [8].

There is a growing literature around the role of the arts in helping us regulate and adapt our responses to experiences and information [9] and an emerging literature on the arts as an entry point to develop data literacy [10]. STEAM (Science, Technology, Engineering, Arts, and Mathematics) education combines the arts with STEM subjects to increase creativity, critical thinking and problem-solving skills [11]. It is also used to engage students traditionally absent from STEM e.g. female students and to engage a variety of types of learners [12]. In the context of data literacy, Bhargava et al. used an arts-based intervention in socially divided communities in Brazil to help people feel less intimidated by data, think critically about the data gathering process, and generate a sustained interest in using data for storytelling and creating social change [2]. Data 'sculptures' i.e. physical manifestations of data, have also been used to engage novice learners with data and build self-efficacy [13]. Matuk et al. integrated arts and mathematics to develop middle school students' data literacy through the use of a data drawing and data sculpture [14]. The goal was to encourage students to connect data with context, identify patterns and argue with data. DesPortes et al. reported on a co-designed data-dance unit where middle school students communicated graphical representations of data through dance to develop their sense making of data [15].

Despite the growing evidence of research in this area, to our knowledge, there has been no comprehensive review to date on the nature and scope of publications using the arts to enhance data literacy. This proposed scoping review aims to address this gap. Scoping reviews are a relatively recent method for synthesising research evidence. They involve similar approaches to other review processes (the use of transparent and clearly articulated search methods to collate, screen, summarize and interpret a body of literature) but differ in intent. The key purpose of a scoping review is to identify and map the available evidence in the field [16]. Scoping reviews are useful for mapping an emerging body of literature by location, time, discipline, context and study design, and can also identify gaps in the literature. Given the interdisciplinary and broad aim of this review, we have identified a scoping review as the most useful method for establishing the current state of the field on the role of the arts to enhance data literacy. This protocol outlines the process we will use to identify and map relevant literature.

The strategy developed in this protocol focuses on three key areas. The first of these is to identify which art forms (or combination of forms) are being reported in the published literature. There is existing evidence that visual art forms, particularly photography, are the most commonly used media in arts-based research generally [17] with less evidence for music, for example [18], however, we do not currently know whether this trend also exists in the use of the arts in data literacy projects.

A second key area of interest for this protocol is to identify which population groups and settings are represented in published work around the arts and data literacy. Specifically, we wish to enhance knowledge related to characteristics such as age profile (is it primarily used in the context of young person education for example?), gender, ethnicity and social group (are groups with limited access to data more or less represented in the literature?) and setting (e.g. is it primarily used in school settings, health settings or in the community?).

A third key area driving the review strategy is to develop enhanced understanding of the rationale for using the arts in the context of data literacy (e.g. is it used to increase accessibility, engagement, or develop critical thinking skills?).

This review will be carried out by a team of interdisciplinary scholars including artists and data scientists. We have identified the following key objectives:

- Identify which art forms (or combination of forms) are being reported in the published literature.

- Identify which population groups and settings are represented in the published literature.

- Analyze the findings to increase understanding of the rationale for the use of the arts to enhance data literacy.

Table 1 presents definitions and inclusion/exclusion criteria related to the role of the arts in data literacy.

## Materials and methods

This protocol has been reported using the Preferred Reporting Items for Systematic Reviews and Meta-analyses extension for systematic review protocols (PRISMA-P) [19] S1 Checklist.

### Methodological framework

The proposed scoping review will utilize Arksey and O'Malley's methodological framework which recommends five non-optional stages and an optional sixth stage: (i) Identifying the research question; (ii) Identifying relevant studies; (iii) Study selection; (iv) Charting the data; (v) Collating, summarizing, and reporting results; and (vi) Consultation [20]. As proposed by Levac et al., this scoping review will include the sixth 'consultation' phase as a necessary step in order to include the expertise of stakeholders in discussions around findings, recommendations for research and practice, and dissemination plans [21].

**Table 1. Concepts, definitions and boundaries of the research question.**

| Concepts | Associated Definitions | Parameters for Inclusion & Exclusion |
|---|---|---|
| Data Literacy | Data literacy has been defined as "the ability to read, work with, analyze and argue with data" (p.198) [2] | Any research approach that integrates data literacy with the arts is included. |
| | | Research approaches that do not integrate data literacy with the arts are excluded. |
| The Arts | "While the arts have always been conceptually difficult to define, there are a number of cross-cultural characteristics recognized as fundamental to art. These include the art object (whether physical or experiential) being valued in its own right rather than merely as a utility; providing imaginative experiences for both the producer and audience; and comprising or provoking an emotional response. In addition, the production of art is characterized by requiring novelty, creativity or originality; requiring specialized skills; and relating to the rules of form, composition or expression (either conforming or diverging)" (p.1) [9]. | Studies that include a broad range of art forms are included. |
| | | Studies that focus on the use of the arts in the context of data literacy are included. |
| | The Arts Act, 2003, (Ireland) defines the arts as: | Studies that do not focus on the use of the arts in the context of data literacy are excluded. |
| | "any creative or interpretative expression (whether traditional or contemporary) in whatever form, and includes [visual arts, theatre, literature, music, dance, opera, film, circus and architecture". | |
| | http://www.irishstatutebook.ie/eli/2003/act/24/enacted/en/html | |

## Stage 1: Identifying the research question

We used the Population Concept Context framework to develop our review question. The population is all human populations. The concept is the role of the arts and the context is data literacy. In line with the exploratory purpose of scoping reviews, we identified one key research question: What is the role of the arts in enhancing data literacy? Our key aim is to identify and map the available literature on the role of the arts in enhancing data literacy.

*Eligibility criteria*. Any research approach that uses the arts in the context of data literacy is included. Studies dating from 2002 to the commencement of the search are included. The time period reflects the relatively recent use of the term 'data literacy' [1]. Only studies in the English language are included. All population groups are included. All quantitative, qualitative, and mixed methods studies, reviews, books and book chapters, text and opinion papers, and grey literature that discuss the arts in the context of data literacy will be eligible for inclusion, as outlined in Table 2.

## Stage 2: Identifying relevant studies

*Search strategy*. All members of the interdisciplinary research team were involved in designing and developing the search strategy for the proposed scoping review. Multiple discussions led to collective agreement around the most appropriate definitions, concepts and terminology relating to our research question and objectives. Terms related to data literacy and the arts are given in Table 3. These will be used across all databases encompassing the disciplines of arts and humanities, social science, education, science and technology.

Preliminary searches were conducted in Scopus and Web of Science in title, abstract, key-words and subject heading fields. The search strategy will be adapted to suit each database and further refined in collaboration with a faculty librarian. The research team will meet regularly during the review process to discuss the efficacy of the search strategy, and any required refinements [20]. Comprehensive searches will be carried out in the following databases: ERIC, Academic Search Complete, Social Sciences Citation Index, Sociological Abstracts, Web of Science Core Collection, Scopus, ProQuest Dissertations & Theses A& I, and Google Scholar. This list may be refined due to the iterative nature of the review process. The reference lists of all included sources of evidence will be screened for additional potentially relevant studies. Any changes in the search strategy will be documented and reported in the results paper. Table 4

**Table 2. Eligibility criteria.**

| Review Category | Inclusion Criteria | Exclusion Criteria | Rationale |
|---|---|---|---|
| **Population** | Any human study population | None | To conduct a broad search that includes any study population |
| **Language** | Written in English | Written in any language other than English | Reviewers' first language is English |
| **Time Period** | 2002 to present | Any publication outside of these dates | For currency and to reflect the recent use of the term 'data literacy' |
| **Study Focus** | Publications on the use of the arts in the context of data literacy | Publications that do not discuss the use of the arts in the context of data literacy | To build an evidence base around the use of the arts to enhance data literacy |
| **Types of Sources** | Journal articles, reviews, books, book chapters, text and opinion papers, web resources—grey literature (e.g. conference proceedings, theses/dissertations, reports). | Book reviews | Scoping studies can incorporate multiple study designs in either published or grey literature. The aim is to capture a comprehensive body of literature in the exploration of a broad research question. |
| **Study Location** | Any study location | None | Data literacy is a concept used globally |

**Table 3. Terms related to data literacy and the arts.**

| |
|---|
| "Data literacy" |
| "Statistical literacy" |
| Arts including "literary arts", "visual arts", "performing arts", "oral narrative arts", "digital arts", "electronic arts", "online arts" |
| poem |
| poetry |
| "creative writing" |
| novel* |
| story* including storytelling |
| stories |
| photography |
| film including "documentary film" |
| video |
| drama |
| theatre including "video-recorded theatre" |
| theater |
| puppetry |
| drawing |
| collage |
| painting |
| graffiti |
| textile |
| mosaic* |
| masks |
| artefact |
| artifact |
| sculpture |
| singing |
| song* |
| music* including "musical instrument" |
| danc* |
| "live art" |
| "body art" |
| "performance art" |
| STEAM |
| SciArt |
| ArtScience |

provides examples of preliminary database searches conducted in Scopus and Web of Science Core Collection.

## Stage 3: Study selection

Citations identified in the search will be compiled and exported to Endnote 20 (Clarivate Analytics, PA, USA) bibliographic software, and duplicates will be removed. The screening process, by title and abstract, will be independently piloted by two reviewers on a subset of citations and then all citations will be screened, guided by the inclusion and exclusion criteria. Full-texts of potentially relevant citations will be reviewed by two independent reviewers with disagreement resolved by discussion or by a third reviewer. Reasons for the exclusion of

**Table 4. Examples of preliminary database searches.**

| |
|---|
| Database: Scopus (searching 'title, abstract and keywords' field) |
| Date: 10/01/23 |
| ( TITLE-ABS-KEY ( "data literacy" OR "statistical literacy" ) AND TITLE-ABS-KEY ( arts OR poem OR poetry OR "creative writing" |
| OR novel* OR story* OR stories OR photography OR film OR video OR |
| drawing OR collage OR painting OR graffiti OR textile* OR |
| mosaic* OR masks OR artefact OR artifact OR sculpture OR singing OR song* |
| OR music* OR danc* OR drama OR theatre OR theater OR puppetry OR "live art" OR "body art" OR "performance art" |
| OR STEAM OR ArtScience OR SciArt) ) AND PUBYEAR > 2001 AND ( LIMIT-TO ( LANGUAGE , "English" ) ) |
| Results: 152 documents |
| Database: Web of Science Core Collection (searching 'topic' field: title, abstract, author keywords, and Keywords Plus). |
| ("data literacy" OR "statistical literacy") AND (arts OR poem OR poetry OR "creative writing" OR novel* OR story* OR stories OR |
| photography OR film OR video OR drawing |
| OR collage OR painting OR graffiti OR textile* OR mosaic* OR masks OR |
| artefact OR artifact OR sculpture OR singing OR song* OR |
| music* OR danc* OR drama OR theatre OR |
| theater OR puppetry OR "live art" OR "body art" OR "performance art" OR STEAM OR ArtScience OR SciArt) (Topic) and English (Languages) |
| Timespan: 2002-01-01 to 2023-01-10 (Publication Date). |
| Results: 121 documents |

publications will be documented. These processes and results will be presented in a PRISMA extension for scoping reviews (PRISMA-ScR) flow diagram [22]. We will use the PRISMA-ScR checklist to ensure adherence to best practice methods for conducting a scoping review.

## Stage 4: Charting the data

Arksey and O'Malley note that the application of "a common analytical framework" through the collection of "standard information on each study" (p. 16) can yield more useful information than merely providing a summary of the identified studies [20]. To this end, we will use a chart organized in MS Excel for the data extraction process (Table 5). Standard recommended charting elements and questions relevant to this review will guide the charting process [23]. The chart includes additional rows in order to record how articles address the core concept of this review i.e. the role of the arts, in the context of data literacy. These additional rows will provide a foundation for the next stage of the review process, i.e. analysis of the data, reporting the results, and applying meaning to the review results [21]. The data extraction form will be trialed independently by two authors on the first ten included studies who will meet to ensure that the data extraction table accurately aligns with the key purpose of the review, and to discuss and agree on any necessary adaptations [21]. Table 5 outlines preliminary charting elements and questions which will guide the data extraction process.

## Stage 5: Collating, summarizing and reporting the results

While incorporating rigorous and transparent methods, scoping reviews differ from systematic reviews in that they are not required to include a critical appraisal of the sources of evidence [16, 20, 21]. We will analyze the evidence using descriptive numerical summaries and qualitative thematic analysis [22,24]. A descriptive numerical summary will outline the main

**Table 5. Preliminary charting elements and questions (adapted from [23, 25]).**

| Categories | Questions |
|---|---|
| **Publication Details** | |
| Author(s) | Who are the authors of the publication? |
| Year of Publication | When was the paper/study published? |
| Country of origin | Where was the study carried out? |
| Publication Type | Is the publication a journal article, book or book chapter, review, opinion paper, grey literature, other? |
| **General Overview of Study** | |
| Aims or Purpose | What were the aims or purpose of the study? |
| | What was the rationale for using the arts? |
| Methodology | What methodological design was utilized for the study? |
| **Key findings relating to the role of the arts in data literacy** | |
| Methods | What specific methods (e.g. qualitative, quantitative, mixed methods) were utilized? |
| | Which art forms were used? |
| | Was data literacy defined and if so, how? |
| | Was data literacy measured and if so, how? |
| | Was there an evaluation of the role of the arts in enhancing data literacy and if so, how? |
| **Study Population** | What population groups are being studied in the literature? |
| | What was the sample size? |
| **Research Setting** | Was it a community-based setting? |
| | Was it an educational setting? |
| **Findings/results** | How have the arts been used to enhance data literacy? |
| | Was the evaluation of the role of the arts positive, negative or mixed? |

characteristics of the included studies such as year of publication, country of origin, methodological design, study population and sample size, and art form used. In addition, a thematic summary will provide an overview of the literature on the use of the arts in the context of data literacy in line with our research question and key objectives [23]. Findings will be organized into thematic categories relating to research methods, key findings and research gaps. As recommended by Peters et al. [23], we will present the results in the context of the overall study purpose to describe and map the available evidence for the role of the arts in enhancing data literacy, and discuss implications in the broader context of future research and practice [21].

## Stage 6: Consultation

As recommended by Levac et al., we will include consultation with stakeholders as a necessary component of the proposed scoping review [21]. This scoping review represents the work of the PART-IM (Participatory and Arts-Based Methods for Involving Migrants in Health Research) research cluster which brings together a multi-disciplinary group of scholars including the performing arts and data science. They will be invited to discuss and evaluate preliminary findings, and their perspectives will help to inform ideas around future research in this area [21]. We will also use the consultation stage to develop effective dissemination strategies with stakeholders in the field.

## Discussion

Publishing a review protocol is increasingly considered best practice in evidence synthesis to increase transparency, reduce duplication and avoid publication bias [26]. It also facilitates

assessment and feedback from the research community in advance of the review being conducted. This protocol outlines the process we will use to identify and map the available evidence for the role of the arts in enhancing data literacy. It has been informed by preliminary searches, discussions and joint decisions in an interdisciplinary team and will guide the conduct and reporting of the review. We acknowledge that the search for studies in English language only is a limitation. We also acknowledge that this search is limited to academic products and may not capture all relevant outputs in this interdisciplinary space with the arts. The scoping review will, however, be an important step to describe and map the available academic evidence for the role of the arts in enhancing data literacy.

## Supporting information

**S1 Checklist. PRISMA-P checklist.**
(DOC)

## Author Contributions

**Conceptualization:** Ailish Hannigan, Helen Phelan.

**Funding acquisition:** Ailish Hannigan, Helen Phelan.

**Methodology:** Ailish Hannigan, Fran Garry, Conor Byrne, Helen Phelan.

**Resources:** Ailish Hannigan.

**Supervision:** Ailish Hannigan, Helen Phelan.

**Writing – original draft:** Ailish Hannigan, Fran Garry, Helen Phelan.

**Writing – review & editing:** Ailish Hannigan, Fran Garry, Conor Byrne, Helen Phelan.

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
