## [Decision Letter · Decision Letter 0]

1 Dec 2022

PONE-D-22-27966The role of the arts in enhancing data literacy: a scoping review protocolPLOS ONE

Dear  Dr. Hannigan,

Thank you for submitting your manuscript to PLOS ONE. After careful consideration, we feel that it has merit but does not fully meet PLOS ONE’s publication criteria as it currently stands. Therefore, we invite you to submit a revised version of the manuscript that addresses the points raised during the review process.

We look forward to receiving your revised manuscript.

Kind regards,

Marcos André André Vannier-Santos, Ph.D.

Academic Editor

PLOS ONE

Journal Requirements:

"This work has been supported by the Irish Research Council https://research.ie/ under grant number COALESCE/2022/1664 (HP, AH)."

Reviewers' comments:

Reviewer's Responses to Questions

**Comments to the Author**

1. Does the manuscript provide a valid rationale for the proposed study, with clearly identified and justified research questions?

Reviewer #1: Yes

Reviewer #2: Yes

2. Is the protocol technically sound and planned in a manner that will lead to a meaningful outcome and allow testing the stated hypotheses?

Reviewer #1: Yes

Reviewer #2: Yes

3. Is the methodology feasible and described in sufficient detail to allow the work to be replicable?

Reviewer #1: Yes

Reviewer #2: Yes

4. Have the authors described where all data underlying the findings will be made available when the study is complete?

Reviewer #1: Yes

Reviewer #2: Yes

5. Is the manuscript presented in an intelligible fashion and written in standard English?

Reviewer #1: Yes

Reviewer #2: Yes

6. Review Comments to the Author

You may also provide optional suggestions and comments to authors that they might find helpful in planning their study.

Reviewer #1: This submission outlines a scoping review protocol intended to synthesize academic understanding of how, when, and where the arts is being used to enhance data literacy. The methods are sound in relation to other reviews I have read or taken part in. The resulting review would be a valuable contribution to the growing body of work on arts and data literacy in multiple settings.

Introduction:

The review of rationale and context is clear and easy to understand for readers. The citations included are contextually appropriately. I have concerns about the scope being limited to academic products, because my personal experience in the field has shown numerous examples in k-12 learning settings that aren't producing formal academic outputs, and data art pieces in gallery settings. This is a well-known concern in mixed space reviews such as this one, so I encourage the authors to engage it and acknowledge it as a limitation to the methods.

Materials and Methods

I am unfamiliar with scoping reviews, but they are summarized effectively for me. The queries in Table 3 needs some work - there are multiple examples of overlapping clauses (if I've understood the boolean syntax correctly). For instance, once "arts" is included there is no need to include "literary arts", "visual arts", etc. And "danc*" likely matches "dance" so that is redundant; same for "music*" and "musical instrument". In addition some clauses should be more generic, such as using "mosaic*" instead of "mosaics". These detailed comments should be taken in context of the search platform, whose syntax I am not an expert on (ie. does it auto-stem? match partial words? etc). I also might suggest including the phrase "performance", to capture the telling of a data story though arts means.

Other comments:

The process will necessitate some hard-to-describe subjective judgements on inclusion/exclusion. This will make replication difficult, but the dual-review nature described should allow for transparency on application of the criteria.

Reviewer #2: This manuscript outlines a protocol to guide a scoping review aiming to identify and map the available evidence for the role of the arts in enhancing data literacy. It is a very interesting topic and, in my opinion, the protocol is well conceived and proposed, deserving publication,

I have concerns on the abstract and the text (discussion), as follows:

Abstract: should indicate more precisely what has been done. It states “ to date, there has been no comprehensive review of publications on the role of the arts in the context of data literacy” and it follows… “The key aim of this proposed scoping review is to identify and map the available evidence for the role of the arts in enhancing data literacy.” The reader concludes that the paper will perform the scoping review. However, in the next sentence we discover that the paper proposes a protocol to perform the scoping review on this subject. The frustration of reading a paper with an expectation and not finding the final review should be mitigated with a starting statement in the abstract clearly showing its goal (my suggestion): “This paper presents a protocol and a search strategy framework to perform a scoping review to identify and map the available evidence for the role of the arts in enhancing data literacy”. PLOS ONE welcomes original research submissions from the natural sciences, medical research, engineering, as well as the related social sciences and humanities, including (…) Protocols (…) that describe detailed plans for research projects. It is the case in the present manuscript, that fits PLOS ONE scope.

The text:

The authors adequately justify the need of the scoping review, present 4 Tables describing the concepts, definitions, and boundaries of the research question (Table 1), the eligibility criteria (Table 2), two examples of preliminary database searches (Table 3) showing 110 documents retrieved using the proposed strategy search in Scopus database, and 98 documents in Web of Science Core Collection, and preliminary charting elements and questions (Table 4) to apply in the documents found. The result is the proposed protocol as a methodological framework, which is interesting. Discussion could comment on some limitations of the study protocol. I can indicate at least 3:

(1) the use of a short ten-year period (>2012 publications), justified as “for currency”, but incompatible with the date of year of the Ottawa charter for health promotion (1986); the Fancourt and Finn scoping review published by WHO in 2019 relating the role of the arts and health cited in ref #9 presents 962 references, and the choice of starting this new scoping review at 2012 is not obvious nor necessarily adequate.

(2) the option for including only publications written in English: given that authors will search other sources that English-written databases (ERIC, Scholar google, grey literature, etc), it is a limitation to exclude, at least, French, Spanish or Portuguese written documents.

(3) the absence of the keyword ArtScience or SciArt, since STEAM is an acronym that does not represent all emerging literature on the arts as an entry point to science evidence.

Independently of these concerns, my overall opinion on the paper is that it deserves publication due to the originality and relevance of the subject, and adequacy of methodology.

7. PLOS authors have the option to publish the peer review history of their article (what does this mean?). If published, this will include your full peer review and any attached files.

Reviewer #1: No

Reviewer #2: **Yes: **TANIA C. ARAUJO-JORGE

---

## [Author Response · Author response to Decision Letter 0]

13 Jan 2023

We thank the reviewers for their positive and constructive feedback, which is particularly useful at this protocol stage of our project. We have addressed their comments on a point to point basis below.

Reviewer #1: This submission outlines a scoping review protocol intended to synthesize academic understanding of how, when, and where the arts is being used to enhance data literacy. The methods are sound in relation to other reviews I have read or taken part in. The resulting review would be a valuable contribution to the growing body of work on arts and data literacy in multiple settings.

Response of the authors: Thank you for the positive feedback on our review protocol.

Introduction:

The review of rationale and context is clear and easy to understand for readers. The citations included are contextually appropriately. I have concerns about the scope being limited to academic products, because my personal experience in the field has shown numerous examples in k-12 learning settings that aren't producing formal academic outputs, and data art pieces in gallery settings. This is a well-known concern in mixed space reviews such as this one, so I encourage the authors to engage it and acknowledge it as a limitation to the methods.

Response of the authors: We agree that this is a limitation of our review and have now added a sentence acknowledging this in the Discussion - ‘We acknowledge that this search is limited to academic products and may not capture all relevant outputs in this interdisciplinary space with the arts.’

Materials and Methods

I am unfamiliar with scoping reviews, but they are summarized effectively for me. The queries in Table 3 needs some work - there are multiple examples of overlapping clauses (if I've understood the boolean syntax correctly). For instance, once "arts" is included there is no need to include "literary arts", "visual arts", etc. And "danc*" likely matches "dance" so that is redundant; same for "music*" and "musical instrument". In addition some clauses should be more generic, such as using "mosaic*" instead of "mosaics". These detailed comments should be taken in context of the search platform, whose syntax I am not an expert on (ie. does it auto-stem? match partial words? etc). I also might suggest including the phrase "performance", to capture the telling of a data story though arts means.

Response of the authors: Thank you for the feedback on the search terms. We have included a new Table 3 with all terms of interest. We have also revised the search terms for the two databases in Table 4, removing overlapping terms and using wildcards where appropriate. We have added “performance art” as a search term. Other types of performance will be captured using the search terms e.g. story, arts, theatre. The preliminary searches have been re-run with the expanded date range (see response to reviewer 2) and the changes in search terms. The results are given in Table 4. 

Other comments:

The process will necessitate some hard-to-describe subjective judgements on inclusion/exclusion. This will make replication difficult, but the dual-review nature described should allow for transparency on application of the criteria.

Response of the authors: We agree with the importance of transparency on applying the criteria and that the dual-review will help with this. We will also document reasons for exclusion during the process at the full-text screening stage.

Reviewer #2: This manuscript outlines a protocol to guide a scoping review aiming to identify and map the available evidence for the role of the arts in enhancing data literacy. It is a very interesting topic and, in my opinion, the protocol is well conceived and proposed, deserving publication.

Response of the authors: Thank you for the positive feedback on our review protocol.

I have concerns on the abstract and the text (discussion), as follows:

Abstract: should indicate more precisely what has been done. It states “ to date, there has been no comprehensive review of publications on the role of the arts in the context of data literacy” and it follows… “The key aim of this proposed scoping review is to identify and map the available evidence for the role of the arts in enhancing data literacy.” The reader concludes that the paper will perform the scoping review. However, in the next sentence we discover that the paper proposes a protocol to perform the scoping review on this subject. The frustration of reading a paper with an expectation and not finding the final review should be mitigated with a starting statement in the abstract clearly showing its goal (my suggestion): “This paper presents a protocol and a search strategy framework to perform a scoping review to identify and map the available evidence for the role of the arts in enhancing data literacy”. PLOS ONE welcomes original research submissions from the natural sciences, medical research, engineering, as well as the related social sciences and humanities, including (…) Protocols (…) that describe detailed plans for research projects. It is the case in the present manuscript, that fits PLOS ONE scope.

Response of the authors: We acknowledge that the abstract could have been clearer that this is a protocol and have now amended it to include the sentence suggested by the reviewer with the addition of methodological framework – “This paper presents a protocol and a methodological framework to perform a scoping review to identify and map the available evidence for the role of the arts in enhancing data literacy.”

The text:

The authors adequately justify the need of the scoping review, present 4 Tables describing the concepts, definitions, and boundaries of the research question (Table 1), the eligibility criteria (Table 2), two examples of preliminary database searches (Table 3) showing 110 documents retrieved using the proposed strategy search in Scopus database, and 98 documents in Web of Science Core Collection, and preliminary charting elements and questions (Table 4) to apply in the documents found. The result is the proposed protocol as a methodological framework, which is interesting. 

Response of the authors: Thank you for the positive feedback on our methodological framework.

Discussion could comment on some limitations of the study protocol. I can indicate at least 3:

(1) the use of a short ten-year period (>2012 publications), justified as “for currency”, but incompatible with the date of year of the Ottawa charter for health promotion (1986); the Fancourt and Finn scoping review published by WHO in 2019 relating the role of the arts and health cited in ref #9 presents 962 references, and the choice of starting this new scoping review at 2012 is not obvious nor necessarily adequate.

Response of the authors: We acknowledge that the starting point for the review (2012) is arbitrary. We have now extended our search from 2002 to date (20 years) and have justified this by the use of relatively recent used of the term ‘data literacy’ which is rarely used before 2002 (we have found no references pre-2002 in preliminary searches). The preliminary searches have been re-run with the expanded date range and the changes in search terms. The results are given in Table 4. 

(2) the option for including only publications written in English: given that authors will search other sources that English-written databases (ERIC, Scholar google, grey literature, etc), it is a limitation to exclude, at least, French, Spanish or Portuguese written documents.

Response of the authors: we acknowledge English language only is a limitation of this review (constrained by the first language of the authors) and have now acknowledged this in the discussion.

(3) the absence of the keyword ArtScience or SciArt, since STEAM is an acronym that does not represent all emerging literature on the arts as an entry point to science evidence.

Response of the authors: Thank you for this important point. We have now added ArtScience and SciArt as potential terms for the search strategy in addition to STEAM in a new Table 3 and a revised search strategy in Table 4.

Independently of these concerns, my overall opinion on the paper is that it deserves publication due to the originality and relevance of the subject, and adequacy of methodology.

Response of the authors: Thank you for the positive feedback on our paper.

Response to academic editor

Journal Requirements:

Response of the authors: we have checked that the style of our manuscript follows the guidelines and used the requested file names.

"This work has been supported by the Irish Research Council https://research.ie/ under grant number COALESCE/2022/1664 (HP, AH)."

Response of the authors: we have amended the funding statement to "This work has been supported by the Irish Research Council https://research.ie/ under grant number COALESCE/2022/1664 (HP, AH). There was no additional external funding received for this study”.

Response of the authors: We have included a caption for the supporting information file at the end of our manuscript and updated the in-text citation.

Response of the authors: We have checked our reference list. There are no changes.

---

## [Decision Letter · Decision Letter 1]

1 Feb 2023

The role of the arts in enhancing data literacy: a scoping review protocol

PONE-D-22-27966R1

Dear Dr. Hannigan,

We’re pleased to inform you that your manuscript has been judged scientifically suitable for publication and will be formally accepted for publication once it meets all outstanding technical requirements.

Kind regards,

Elias Garcia-Pelegrin, Ph.D

Academic Editor

PLOS ONE

Additional Editor Comments (optional):

The authors have addressed all the reviewer comments adequately, I am very pleased to be able to recommend acceptance at PLOS ONE.

Reviewers' comments:

Reviewer's Responses to Questions

**Comments to the Author**

1. Does the manuscript provide a valid rationale for the proposed study, with clearly identified and justified research questions?

Reviewer #2: Yes

2. Is the protocol technically sound and planned in a manner that will lead to a meaningful outcome and allow testing the stated hypotheses?

Reviewer #2: Yes

3. Is the methodology feasible and described in sufficient detail to allow the work to be replicable?

Reviewer #2: Yes

4. Have the authors described where all data underlying the findings will be made available when the study is complete?

Reviewer #2: Yes

5. Is the manuscript presented in an intelligible fashion and written in standard English?

Reviewer #2: Yes

6. Review Comments to the Author

You may also provide optional suggestions and comments to authors that they might find helpful in planning their study.

Reviewer #2: The authors have accepted and correctly addressed all the points that the reviewers raised upon the first version. The paper deserves publication

7. PLOS authors have the option to publish the peer review history of their article (what does this mean?). If published, this will include your full peer review and any attached files.

Reviewer #2: **Yes: **TANIA C. ARAUJO JORGE

---

## [Editor Report · Acceptance letter]

3 Feb 2023

PONE-D-22-27966R1 

The role of the arts in enhancing data literacy: a scoping review protocol 

Dear Dr. Hannigan:

I'm pleased to inform you that your manuscript has been deemed suitable for publication in PLOS ONE. Congratulations! Your manuscript is now with our production department. 

Kind regards, 

on behalf of

Dr. Elias Garcia-Pelegrin 

Academic Editor

PLOS ONE